

# The response of seagrass (*Posidonia oceanica*) meadow metabolism to CO₂ levels and hydrodynamic exchange determined with aquatic eddy covariance

Dirk Koopmans[1], Moritz Holtappels[2], Arjun Chennu[1], Miriam Weber[3], and Dirk de Beer[1]

[1]Max Planck Institute for Marine Microbiology, Bremen, 28359, Germany
[2]Alfred Wegener Institute, Helmholtz Center for Polar and Marine Research, Bremerhaven, 27515, Germany
[3]HYDRA Institute for Marine Sciences, Elba Field Station, Campo nell'Elba, 57034, Italy

*Correspondence to*: Dirk Koopmans (dkoopman@mpi-bremen.de)

**Abstract.** We investigated light, water velocity, and CO₂ as drivers of primary production in Mediterranean seagrass (*Posidonia oceanica*) meadows and neighboring bare sands using the aquatic eddy covariance technique. Study locations included an open-water meadow and a nearshore meadow, the nearshore meadow being exposed to greater hydrodynamic exchange. A third meadow was located at a CO₂ vent. We found that, despite the oligotrophic environment, the meadows had a remarkably high metabolic activity, up to 20 times higher than the surrounding sands. They were strongly autotrophic, with net production half of gross primary production. Thus, *P. oceanica* meadows are oases of productivity in an unproductive environment. Secondly, we found that turbulent oxygen fluxes above the meadow can be significantly higher in the afternoon than in the morning at the same light levels. This hysteresis can be explained by the replenishment of nighttime-depleted oxygen within the meadow during the morning. Oxygen depletion and replenishment within the meadow do not contribute to turbulent O₂ flux. The hysteresis disappeared when fluxes were corrected for the O₂ storage within the meadow and, consequently, accurate metabolic rate measurements require measurements of meadow oxygen content. We further argue that oxygen-depleted waters in the meadow provide a source of CO₂ and inorganic nutrients for fixation, especially in the morning. Contrary to expectation, meadow metabolic activity at the CO₂ vent was lower than at the other sites, with negligible net primary production.

## 1 Introduction

Seagrass meadows retain suspended sediments (Fonseca and Fisher, 1986), provide habitat for fish and invertebrates (Heck and Wetstone, 1977; Jenkins and Wheatley, 1998), and supplement food webs in neighboring environments (Fry and Parker, 1979). Seagrass meadows are highly productive (Zieman and Wetzel, 1980; Bay 1984; Frankignoulle and Bouquegneau, 1987) with net production commonly exceeding 1 kg dry weight m⁻² y⁻¹ (Duarte and Chiscano, 1999). Seagrass productivity for a given meadow is determined by the balance of photosynthesis and respiration. These are driven by irradiance (Dennison and Alberte, 1982; Peralta et al., 2002; Ralph et al., 2007) nutrient availability (Short, 1987; Powell et al., 1989; Udy and Dennison, 1987), and temperature (Bulthuis, 1987; Alcoverro et al., 1995; Collier and Waycott, 2014). These responses are modified by water velocity (Fonseca and Kenworthy, 1987; Thomas et al., 2000; Peralta et al., 2006), and CO₂ availability (Koch et al., 1994; Zimmerman et al., 1997). A challenge of seagrass research is quantifying the response to these drivers in seagrass meadows. The attenuation of light and flow in meadows creates steep gradients in irradiance (Dalla Via et al., 1998; Enríquez and Pantoja-Reyes, 2005), turbulent mixing (Ackerman and Okubo, 1993; Koch and Gust, 1999), and the concentrations of biologically-active solutes (Frankignoulle and Distéche, 1984; Semesi et al., 2009). These conditions may alter the response of seagrass meadows to drivers, leading to different observations than in seagrass incubations.





Self-shading in dense meadows may enhance the importance of irradiance as a driver of primary production. For example, in *Zostera marina* leaf incubations, saturating light intensities occur at photon flux densities of $100 - 230$ µmol photon $m^{-2}$ $s^{-1}$ (Drew, 1979; Dennison and Alberte, 1985). Binzer et al., (2006), however, found that measurements on isolated leaves give a false impression of light saturation of canopy photosynthesis. They consistently observed light-limitation in canopies, even at peak

summertime irradiances. As an explanation, the high degree of self-shading in a canopy can create an environment in which a part of the leaves is almost never light saturated (Sand-Jensen et al., 2007). Indeed, in mature *Zostera marina* meadows in summer, in situ light conditions do not typically saturate primary production (Rheuban et al., 2014a). Instead, seagrass meadow photosynthetic production increased linearly to photon flux densities of over $1000$ µmol photons $m^{-2}$ $s^{-1}$.

Meadow attenuation of flow enhances the importance of $CO_2$ as a driver of primary production. At current surface ocean

pH at 20°C, $CO_2(aq)$ represents 0.6% ($13$ µmol $L^{-1}$) of dissolved inorganic carbon in seawater. Carbon dioxide is the preferred substrate for Rubisco, the enzyme responsible for carbon fixation (Falkowski and Raven, 1997). Seagrasses maintain productivity in spite of the low availability of $CO_2$ with an $H^+$-driven mechanism for $HCO_3^-$ uptake (Beer and Rehnberg, 1997). Diffusive boundary layers on leaf surfaces commonly limit photosynthetic production due to $CO_2$ depletion and $O_2$ enrichment at the leaf surface (Raven et al., 1985; Koch, 1994; Mass et al., 2010; Enríquez and Rodríguez-Román 2006). Seagrass meadows, by

attenuating flow, may enhance this limitation. Within seagrass meadows, pH may rise and fall by 0.2 pH units or more due to photosynthesis and respiration (Frankignoulle and Bouquegneau, 1987; Hendriks et al., 2014). During photosynthetic production, as pH increases from 7.9 to 8.1, $CO_2(aq)$ concentrations decrease by 70% (Lewis et al., 1998). This pH increase diminished seagrass photosynthesis in incubations (Invers et al., 2001; Palacios and Zimmerman, 2007). In productive bays, seagrass photosynthesis may be reduced by a third due to photorespiration caused by enhanced $O_2$ and depleted $CO_2$ (Buapet et al., 2013).

Forecasts of ocean acidification for the year 2100 are for a drop of up to 0.4 pH units in the surface ocean (Orr et al., 2005), thus tripling the concentrations of $H^+$ and $CO_2$. The reduction in pH will alter seagrass ecosystems. Losses of calcifying epiphytes are likely (Martin et al., 2008; Kroeker et al., 2013), and these are a significant contributor to meadow primary production (Libes, 1986). The effects of acidification will occur gradually and differ by species, life stage, and environmental factors including inter-species interactions (Andersson et al., 2011). $CO_2$ vents serve as a natural laboratory for investigating the effect of $CO_2$

enrichment. Seagrass at $CO_2$ vents have higher densities, biomass, and greater electron transport rates (Hall-Spencer et al., 2008; Fabricius et al., 2011; Takahashi et al., 2015). They may also be adversely affected by $CO_2$ vent eruption (Vizzini et al., 2010). Further studies of seagrass productivity at $CO_2$ vents is warranted.

In situ eddy covariance (Berg et al., 2003) or open water techniques (Odum, 1956; Howarth and Michaels, 2000) can be used to continuously quantify the productivity of seagrass under the complex conditions that occur within meadows. Recent studies

include eddy covariance flux measurements over meadows of *Zostera marina* (Hume et al., 2011; Rheuban et al., 2014a; Rheuban et al., 2014b), *Thalassia testudinum* (Long et al., 2015), and *Zostera noltii* (Lee et al., 2017). Open water measurements of *P. oceanica* productivity confirm the net autotrophy of the ecosystem and reveal a surprising, seven-fold increase in net production due to removal of sloughed seagrass leaves by storms (Champenois and Borges, 2012). Further measurements are required to investigate the interactions of drivers over diel time-scales.

To address these issues, we performed field measurements of the metabolic performance of *Posidonia oceanica* meadows in the Mediterranean Sea as a function of light intensity, flow and $CO_2$ supply. In summer, surface waters of the Mediterranean are highly oligotrophic (Marty et al., 2002). *Posidonia oceanica* develops dense, productive meadows (Ott, 1980; Bay, 1984). We used the non-invasive eddy covariance technique to quantify net $O_2$ fluxes above the seagrass meadows, thus integrating over the complete seagrass ecosystems. Diurnal changes in in-situ primary production were examined over 3 different *Posidonia oceanica*

meadows in the NW Mediterranean Sea. To account for the effects of hydrologic exchange, reference meadows were included at



the same depths in open-water and nearshore environments. The nearshore meadow was exposed to greater water velocities and waves. Waves particularly enhance hydrodynamic mixing within seagrass canopies (Koch and Gust 1999; Hansen and Reidenbach 2017). A meadow at a $CO_2$ vent was included to test the hypothesis that an enhanced $CO_2$ supply enhances seagrass primary production. We also compared seagrass meadow productivity to that of surrounding sands.

## 2 Methods

### 2.1 Study sites

Our primary research site was the island of Elba in the Tuscan Archipelago. The island is surrounded by *Posidonia oceanica* beds from depths of 5 m to 40 m. $O_2$ fluxes were determined over an open-water seagrass meadow in 2016 (15 to 18 May) and a nearshore meadow in 2017 (20 – 24 May). The open-water meadow (42.7421 N, 10.1183 E) was located 300 m from the southwest corner of the island, whereas the nearshore meadow (42.8087 N, 10.1472 E) was only 60 m off the north shore. Both meadows were at 13 m depth and known to persist for over 20 years. Percent areal coverage, estimated from plan-view underwater images, was high (95-99%) at both meadows (Table 1). Water velocities measured 0.3 m above the open-water meadow (mean of 1.2 cm $s^{-1}$) were lower than at the nearshore meadow (mean of 2.6 cm $s^{-1}$; Table 1). Oxygen fluxes were also measured over bare sands adjacent to the nearshore meadow.

A second study site was 100 m off Basiluzzo islet (38.6625 N, 15.1189 E) near Panarea Island in the Aeolian Archipelago. We measured $O_2$ fluxes over a *P. oceanica* meadow (13 m depth) and sediments (17 m depth), where $CO_2$ gas bubbles rise through the seafloor over several 100 $m^2$, enriching surface waters with DIC (Caramanna et al., 2011). Meadow height was half that of the Elba meadows (Table 1). The vent introduced $CO_2$ as well as reduced substances to the water column. This resulted in slightly elevated DIC (2294 ± 34 versus 2220 ± 21 µmol $L^{-1}$) and N-nutrients ($NO_3^-$, $NO_2^-$ and $NH_4^+$; 3.2 ± 3.1 µmol $L^{-1}$ versus 0.73 ± 1.0 µmol $L^{-1}$) at Panarea compared to Elba. At Elba and Panarea phosphates were below detection (0.027 µmol $L^{-1}$). Water temperatures at all sites were also similar (17 – 18 °C).

### 2.2 Flux measurements

The eddy covariance technique utilizes high frequency measurements of solute concentration and water velocity at a single point above the habitat of interest to quantify solute fluxes over 10 $m^2$ footprint located directly upstream (Berg et al., 2003; Berg et al., 2007). Water velocity was determined at 16 Hz with an acoustic Doppler velocimeter (Nortek). Oxygen concentration was determined at 5 Hz with high speed ($t_{90}$ = 0.25s) micro-optode sensors of 50 or 430 µm in diameter (PyroScience). Their illumination, and measurement of their fluorescence, were managed by an oxygen meter (PyroScience) in a submersible housing. Optodes have been successfully used for eddy covariance measurements previously (Chipman et al., 2012; Berg et al., 2016) and they lack stirring sensitivity (Holtappels et al., 2015). Optodes were calibrated in air-saturated and anoxic water before and after each deployment. The instruments were mounted to a tripod frame with narrow legs (4 cm in diameter) spaced 1.2 m apart. The sensors were aligned at a 40-degree angle to the velocimeter, and the tip located 1 cm from the velocimeter measuring volume. The legs of the frame were adjusted to allow for measurements at a height of 0.25 m over sands and 0.3 m above the top of seagrass canopies.

Multi-parameter probes that measured $O_2$ concentration and pH (RBR) were mounted near the top and bottom of the instrument frame. Probes were used to record the vertical $O_2$ concentration gradient in order to identify oxic stratification, which can cause spurious $O_2$ fluxes (Holtappels et al., 2013). The pH sensors were calibrated to NBS buffers. Observed pH values were converted to total scale by subtracting 0.13 pH units, according to Zeebe and Wolf-Gladrow (2001). Irradiance was quantified with

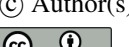



HOBO Pendant sensors (Onset Computer Corp.). Because these luminance sensors are sensitive from 300 – 1200 nm, we cross-calibrated them to quantify the photosynthetically active radiation (PAR, 400-700 nm) using a LI-192 sensor (Li-COR) placed at

the same location underwater (Long et al., 2012). The instrument frame was deployed by divers at the research sites and was aligned to minimize tilt. The time series of mean $O_2$ concentration variation recorded by the fast optodes reproduced those recorded by the $O_2$ logger. Thus, fluctuations recorded by the fast optode were accurate.

### 2.3 Flux calculations

The velocity and oxygen time series, recorded at 16 Hz by the ADV, were downsampled to the optode sampling frequency of 5

Hz and were processed similarly to the procedure described by Holtappels et al., (2013). The tilt of the ADV was corrected using the planar fit method by Wilczak et al., (2001). A running average with a window of 300 s was subtracted from the time series to calculate the fluctuating vertical velocity ($w'$) and $O_2$ concentration ($C'$). Subsequently, the time series of $w'$ and $C'$ were cross-correlated for 1-hour bursts allowing stepwise time shifts to a maximum 4 s (McGinnis et al., 2008). The median time shift was 0.8 s. The time shift with the highest cross correlation coefficient was used to calculate the flux. The mean turbulent $O_2$ flux in the

vertical direction was calculated following Eq. (1):

$$J_w = \overline{w'C'},  \tag{1}$$

where the overbar indicates averaging. Instantaneous fluxes ($w'C'$) were added over time to calculate cumulative fluxes. Particle collision with the tip of $O_2$ minisensors can cause erratic fluxes (e.g., Lorrai et al., 2010). We examined the dataset for spikes in the $O_2$ data that co-occurred with abrupt changes in calculated fluxes. Few hour-long averaging intervals (4%) were affected and

they were excluded from further analyses.

In a further step, we accounted for oxygen storage in the water layer between the measurement volume of the eddy covariance system and the sediment surface. The diurnal variations in mean $O_2$ concentration within this layer do not contribute to oxygen fluxes above the meadow but are nevertheless caused by photosynthesis and respiration. In a first approximation, to account for this $O_2$ storage and quantify biotic production, we assumed that the $O_2$ concentrations at the measurement volume represented

the $O_2$ concentrations in the entire bottom layer. This assumes that there was no vertical oxygen gradient in the bottom layer. We made this correction according to Rheuban et al., (2014b) following Eq. (2):

$$O_2 \ production = EC_{Flux} + h\frac{\Delta C}{\Delta t}  \tag{2}$$

where *EC Flux* is the eddy covariance flux, $h$ is the eddy covariance measuring height above the bed, and $\frac{\Delta C}{\Delta t}$ is the change in the mean $O_2$ concentration at the measurement volume over time for each 1h burst. From these corrected fluxes we infer benthic $O_2$

production and consumption. Negative production represents $O_2$ consumption.

The corrected $O_2$ fluxes, quantified over hourly intervals, were grouped into light and dark (<1% irradiance). These were used to calculate respiration (R), gross primary production (GPP), and net ecosystem metabolism (NEM) in an approach similar to Hume et al., (2011), and reported here in mmol $O_2$ m$^{-2}$ d$^{-1}$. To accommodate datasets that lacked complete daytime or nighttime observations, metabolic fluxes were calculated following Eqs. (3), (4), and (5):

$$R = \left|\overline{Flux_{dark}}\right|  \tag{3}$$



$$GPP = \left(\overline{Flux_{light}} + R\right)\frac{h_{light}}{h_{day}} \qquad (4)$$

$$NEM = (GPP - R) \qquad (5)$$

where $Flux_{dark}$ is a nighttime flux of 1 h duration, $Flux_{light}$ is a daytime flux of 1 h duration, and $h_{light}$ and $h_{day}$ are the durations of the illuminated (14 h) and total length (24 h) of a diurnal cycle. To examine the photosynthetic response of seagrass meadows to

their changing light environment, photosynthesis-irradiance relationships were fit with a hyperbolic tangent function (Jassby and Platt, 1976) which was modified to account for respiration according to Rheuban et al., (2014a). The fit was calculated following Eq. (6):

$$Flux = P_{max} \tanh\frac{I}{I_k} - R_I \qquad (6)$$

where $P_{max}$ is the maximum photosynthetic rate, $I_k$ is the light-saturation parameter, and $R_I$ is respiration. The light-saturation

parameter marks an optimal irradiance above which the quantum yield of photosynthesis substantially declines and photoinhibition may occur (Talling 1957; Falkowski and Raven 1997). The irradiance at which net oxygen production occurs is the irradiance compensation point ($I_C$).

## 2.4 Additional measurements

Supporting measurements were made of nutrient concentrations and the carbonate chemistry in the water column at each of the

study sites. Water samples, collected 1 m above the bed by divers, were filtered over 0.2 µM and then frozen until analysis. Nutrient concentrations ($NH_4^+$, $NO_3^-$, $PO_4^+$) were determined with a QuAAttro nutrient analyzer according to the methods employed by Lichtschlag et al., (2015). Samples for both alkalinity and DIC were collected by divers at the beginning and end of each deployment. Alkalinity samples were kept cool and analyzed by titration on the day of their collection. Samples for DIC analysis were preserved with a saturated mercuric chloride solution (1% by volume), then quantified by flow injection (Hall and Aller,

1992). Time-varying $CO_2(aq)$ concentrations at the $CO_2$ vent were calculated assuming a stable alkalinity and time-varying pH supplied as inputs to the software package CO2SYS (Lewis et al., 1998). Diurnal alkalinity variation caused by seagrass photosynthesis and respiration had a negligible effect on these calculations.

## 3 Results

### 3.1 Oxygen storage in the open-water meadow

Diel cycles of oxygen fluxes were measured over seagrass meadows (Figs. 1, 3, 4, S1). When plotting eddy covariance fluxes versus light intensities in order to obtain photosynthesis-irradiance curves we observed a hysteresis, namely oxygen fluxes were lower in the morning than in the afternoon at the same light intensities (Fig. 2a). For example, $O_2$ flux at 18:00 h was 175 mmol $m^{-2}$ $d^{-1}$ while at the same light intensity in the morning (08:00) it was close to zero. The correction for storage below the measurement volume (Eq. 2) was insufficient to fully reduce this hysteresis. Storage of $O_2$ within *P. oceanica* gas-bearing tissues

could also not explain this pattern. Our laboratory measurements showed that $O_2$ production by the leaves ceased within a few minutes of the cessation of light (data not shown). Instead, we found that the hysteresis in $O_2$ flux at the open-water meadow was caused by $O_2$ storage in the water column within the meadow. The concentration of dissolved $O_2$ within the meadow, at 0.2 m above the bed, differed substantially from the concentration in the eddy covariance measuring volume at 0.3 m above the meadow




(Fig. 1). Thus, there is a mass transfer resistance between the water in- and above the meadow that effectively separates the water

in the meadow from the seawater.

To obtain metabolic rates from measured fluxes, an additional calculation had to be performed to correct for oxygen storage within the open-water meadow. We assumed a linear $O_2$ concentration profile between the measurement volume and an $O_2$ logger placed 0.2 m above the bed. Storage was calculated according Eq. 2, but now from the change over time in the estimated mean $O_2$ concentration below the eddy covariance measurement volume (Fig. S1). Respiration and GPP, calculated from the thus

corrected $O_2$ fluxes (Eqns. 3 & 4), were 1.5 and 1.2 times the measured above-meadow fluxes, respectively. The photosynthesis irradiance curves of corrected fluxes showed a smaller, reversed hysteresis (Fig. 2b). After the correction, the photosynthetic production in the early morning was greater than production in the evening at the same light levels, consistent with the diel photosynthetic production of a *Zostera marina* meadow (Rheuban et al., 2014b). This additional correction for the retention of water within the meadow was not required at the other seagrass meadows where dissolved $O_2$ concentrations in overlying water

matched concentrations within the meadow during the day and at night (data not shown).

### 3.2 Primary production and respiration at Elba

$O_2$ production and consumption, inferred from storage-corrected eddy covariance $O_2$ fluxes, followed similar diurnal patterns at both meadows in Elba (Fig. 3). Daytime $O_2$ production exceeded nighttime $O_2$ consumption at both meadows on both days. As a result, both meadows were net autotrophic (Table 2). Peak GPP was greater in the nearshore meadow, but nighttime $O_2$ uptake was

similar between meadows. The mean NEM of the nearshore meadow exceeded that of the open-water meadow (Table 2). Downwelling PAR was similar in the meadows at Elba and slightly greater over nearshore sands (Fig. 3). The bare sands at Elba were also net autotrophic. Bare sand GPP exceeded R by 40%. However, GPP was 10- to 17-fold smaller than meadow GPP (Table 2). As a result, the NEM of seagrass meadows was up to 20-fold greater than the NEM of sands.

### 3.3 Primary production and respiration at a $CO_2$ vent

At the $CO_2$ vent at the island of Panarea, gas bubbles emerged from sediments at both the seagrass meadow and at the bare sand sites. The rate of gaseous $CO_2$ release increased at low tide, and this increase coincided with a decrease in bottom water pH (Fig. S2). These observations show that gas efflux was affected by tidally-driven expansion of gas reservoirs in the sediments. The pH fell from 8.05 at high tide to a low of 7.55 at low tide, corresponding to a 3.7-fold increase in $CO_2(aq)$ concentration (Lewis et al., 1998). Thus, $CO_2$ was elevated by seepage. Oxygen production followed the same diurnal patterns as the Elba meadows, with

daytime production and nighttime uptake (Fig. 4). Oxygen production by the meadow did not closely follow irradiance. Surprisingly, the GPP of the $CO_2$ vent meadow was less than half of the Elba meadows (Table 2). Mean R at the $CO_2$ vent meadow, however, was similar to the lowest R at Elba. As a result, the meadow was autotrophic but with an NEM five- to ten-fold smaller than the NEM of the Elba meadows. Over neighboring bare sands, the pH of bottom waters dropped from 8.05 at high tide to 7.75 at low tide, corresponding to a doubling of the $CO_2(aq)$ concentration. The GPP of the $CO_2$ vent sands was similar to the GPP of

sands at Elba (Table 2). Respiration of the $CO_2$ vent sands exceeded R in bare sands at Elba. The sands were net autotrophic, but with a small NEM.

### 3.4 Flux-irradiance curves

In all meadows, the diel oxygen flux was a non-linear function of irradiance (Fig. 5), resembling a saturation curve. In none of the meadows $P_{max}$ was reached. The light compensation point ($I_C$) varied between 6 and 17% of the peak irradiance. The light-



saturation parameter ($I_K$) varied was one-third of peak irradiance in the Elba meadows and two-thirds of peak irradiance in the $CO_2$ vent meadow. $P_{max}$ was greatest at the nearshore meadow, followed by the open-water and $CO_2$ vent meadows.

## 4 Discussion

### 4.1 Seagrasses versus sands

The seagrass meadows at Elba and Panarea were 10-20 times as productive as surrounding sands. This is consistent with the high

productivity of *P. oceanica* individuals determined by laboratory-, chamber-, leaf tagging-, or biomass measurements (Zieman, 1974; Ott, 1980; Drew, 1979; Libes, 1986; Pergent et al., 1994), thus emphasizing the importance of seagrass for $CO_2$ sequestration (Fourqurean et al., 2012). Implicit in these prior observations is the high productivity of seagrasses relative to surrounding sands, however, direct comparisons have rarely been made. Previous comparisons were made with benthic chambers, where sand oxygen fluxes were close to the detection limit, and seagrass oxygen fluxes were of the same order as our quantifications (Holmer et al.,

2004; Barrón et al., 2006a). Using DIC mass balance, Gazeau et al., (2005) found smaller primary production than we observed in seagrass and negligible primary production in bare sands.

Meadow productivity in this study was consistent with other open water measurements of *P. oceanica* seagrass meadow productivity. Respiration in *P. oceanica* meadows is much lower than primary production, thus the meadows are highly autotrophic (Champenois and Borges, 2012). Therefore, the decomposition of leaf litter within the ecosystem was substantially smaller than

production. Prior studies utilizing the eddy covariance technique in summer identified greatrer similarity between respiration and primary production in *Zostera marina* meadows in the mid-Atlantic Bight (Rheuban et al., 2014a) and in *Thalassia testudinum* meadows in Florida Bay, USA (Long et al., 2015). In those studies NEM was one-fifth of GPP. Thus, *P. oceanica* meadows may function differently in the highly oligotrophic Mediterranean than *Zostera* and *Thalassia* meadows in a more eutrophic sea. It is of interest why seagrass respiration in the Mediterranean is so much lower than in meadows from Florida or the mid-Atlantic Bight.

Nutrient limitation is an explanation. Nitrogen in pore water can limit the primary productivity of *P. oceanica* meadows in summer (Alcoverro et al., 1997). However, heterotrophic activity is also limited by nutrient availability. The addition of inorganic nitrogen to *P. oceanica* meadow pore water enhances benthic respiration and diminishes the amount of stored organic carbon (Lopez et al., 1998). Because of this, the oligotrophic environment may contribute to the high carbon storage in *P. oceanica* meadows.

In addition to the high productivity of meadows, we found that sands from the Mediterranean are unusually unproductive.

Indeed, R in Elba sands was comparable to that of the continental shelf of the South Atlantic Bight (Jahnke et al., 2000), and among the lowest reported for coastal systems (Huettel et al., 2014). Pelagic primary production is limited in the oligotrophic Mediterranean. Thus, there is little organic carbon to stimulate respiration in this environment. Organic matter concentrations in Mediterranean waters become so low in the summer that many benthic suspension feeders undergo dormancy to avoid starvation (Coma and Ribes, 2003).

We may understand the strong differences in activity between seagrasses and sands in oligotrophic waters in the acquisition and retention of nutrients by seagrass meadows. *P. oceanica* take up nutrients through roots and leaves primarily in winter and early spring, when nutrients are more abundant (Lepoint et al., 2002). Nutrients are reallocated before leaves are shed (Alcoverro et al., 2000). In acquiring nutrients from surrounding waters, the filtering of particles by seagrasses or their epiphytes may be similar to coral reefs (e.g., Rasheed et al., 2002). Seagrasses collect particles passively, through flow-attenuation and

particle deposition (Gacia et al., 1999; Gacia and Duarte, 2001). Seagrass also collect particles actively. Macro- and epifaunal suspension feeders (e.g., hydroids; Fig. S3) enhance particle filtering by orders of magnitude over surrounding sands (Lemmens et al, 1996). Epifauna biomass on *P. oceanica* is one-third of total epiphytic biomass (Lepoint, 1999). Mineralization of captured





particles in seagrass sediment supplies nutrients for seagrass growth (Evrard et al., 2005; Barrón et al., 2006b). Their mineralization by epiphytes may be an additional source of nutrients for seagrass growth, as dissolved nutrients can be retained within the waters

of seagrass meadows (Gobert et al., 2002).

## 4.2 Meadow productivity at the $CO_2$ vent

Despite the seemingly favorable dissolved $CO_2$ availability and the elevated inorganic nitrogen in the water column at Panarea, seagrasses at the vents did not seem to prosper. So, our hypothesis that seagrass growth would be stimulated at the $CO_2$ vent was incorrect. These results contrast with the increased productivity of seagrass at elevated $CO_2$ and reduced pH (Beer and Koch, 1996;

Invers et al., 2001; Palacios and Zimmerman, 2007). Nevertheless, *P. oceanica* is also not stimulated under comparable, intermediate, ocean acidification scenarios at $CO_2$ vents (Hall-Spencer et al., 2008; Cox et al., 2015; Cox et al., 2016). *Thalassia testudinum* is also not stimulated by intermediate $CO_2$ enrichment in mesocosms (Campbell and Fourqurean, 2013). However, seagrass production at the $CO_2$ vent in this study was substantially lower than the productivity of reference meadows. GPP was 40% and R was 70% of that at Elba, and NEM was negligible. The small meadow height (half that of Elba) may contribute to the

lower GPP and R, but the negligible NEM suggests that this meadow was not storing organic carbon. A concern for seagrass meadows exposed to high $CO_2$ levels is the loss of calcifiers due to low pH (Martin and Gattuso, 2009; Donnarumma et al., 2014). Calcifiers may be a significant component of meadow productivity. Calcification offsets photosynthetic $CO_2$ uptake in seagrass meadows by half (Barrón et al., 2006a). However, the community of epibionts at the $CO_2$ vent at Basiluzzo was intact and diverse (Guilini et al., 2017). An alternate explanation is that vent fluids alter the chemistry of the site in a way that limits seagrass

productivity. Seagrass growing near the vent express more stress-related genes than seagrass at reference sites (Lauritano et al., 2015). The vent fluids and sediments at the vents are highly enriched in iron (Price et al., 2015). Iron (III) binds to phosphate and makes it less available in bottom waters (Slomp et al., 1996). Thus, phosphorus may limit seagrass productivity. Vent fluids may also be enriched in potentially harmful trace elements (Vizzini et al., 2013). Generally, the use of natural $CO_2$ vents for studying ocean acidification may require caution. The effects of elevated $CO_2$ may not always be separated from other factors that are

induced by seepage.

In sands, respiration was enhanced at the $CO_2$ vent relative to Elba but GPP was similar. Enhanced $CO_2$ concentrations may enhance the productivity of microalgae and cyanobacteria by alleviating the energetic requirements of their $CO_2$ concentrating mechanisms (Beardall and Giordano, 2002). Benthic diatoms dominated the benthic community at Basiluzzo (Molari et al., 2018). Many benthic diatoms are insensitive to pH changes (Hinga, 2002). However, Johnson et al., (2013) found changes in benthic

diatom assemblages and increases in chlorophyll *a* concentrations at a $CO_2$ vent. Nevertheless, the negligible enhancement of primary production at the $CO_2$ vent is consistent with the small net effects of acidification on benthic diatoms and benthic primary production in other studies (Alsterberg et al., 2013; Fink et al., 2017). Nutrient limitation in oligotrophic environments may limit the response of phytoplankton to elevated $CO_2$ (Gazeau et al., 2017). These results indicate that reduced seawater pH has little net effect on photosynthesis by microphytobenthos.

## 4.3 Hydrodynamic exchange

Reduced nighttime hydrodynamic exchange has ecological benefits for seagrass. The depletion of oxygen at the open-water meadow during the night (Fig. 1) is evidence that hydrodynamic exchange with surrounding waters is limited. Ecologically, the reduced exchange would benefit seagrass if limiting nutrients that were produced during mineralization at night were retained for primary production during the day. To estimate the potential benefit of nutrient retention we assumed Redfield ratios. A 20 $\mu$mol

$L^{-1}$ $O_2$-deficit between the meadow and ambient waters corresponds to a 2.3 $\mu$mol $L^{-1}$ increase in $NO_3^-$. For a 0.6 m-high canopy



this amounts to 1.4 mmol m$^{-2}$ of extra NO$_3^-$, and 12 mmol m$^{-2}$ d$^{-1}$ of GPP. This is between 7 and 9% of GPP of the meadow. Using Redfield ratios for respiration and production, the same increase in GPP applies to phosphorus. These amounts of enrichment are approximately double the observed inorganic nitrogen and phosphorus enrichment near the bed in *P. oceanica* meadows (Gobert et al., 2002). Seagrass are depleted in nitrogen and phosphorus relative to marine seston (Duarte et al., 1990), so the mineralization

of marine seston may stimulate greater GPP. Mineralization would also be a source of CO$_2$ for carbon fixation. Assuming a respiratory quotient of 1, the 20 µmol L$^{-1}$ DIC enrichment corresponds to a small, 0.04 pH decrease (Lewis et al., 1998). This would enhance CO$_2$(aq) by 10%. The potential enrichment of seagrass primary production by nutrient and CO$_2$(aq) retention would be greater for greater excursions in O$_2$ and pH.

        *Posidonia oceanica* may be adapted to diel oscillations in water velocity and waves caused by the solar-driven sea breeze.
For a seagrass meadow to take advantage of nutrient retention, the nutrients would be retained during the night and into the morning. Late in the morning, after nutrients had been depleted, it would be an advantage for hydrodynamic exchange to be enhanced. This would prevent accumulation of O$_2$ which can cause photorespiration (Falkowski and Raven, 1997). This diel pattern in hydrodynamic exchange would be caused by low nighttime water velocities and higher daytime water velocities. Interestingly, this is exactly the diel pattern in water velocities that occurs at the meadow (Fig. 1). The pattern is the result of the solar-driven
sea breeze. This velocity pattern is dominant in the Mediterranean (Azorin-Molina et al., 2011), where tides are almost absent, and thus may be exploited by *P. oceanica*.

### 4.4 Flux-irradiance curves

Our results showed that seagrass meadows are not always light-limited during a typical diel cycle (Fig. 5). Light limitation means that higher light levels will lead to higher photosynthesis rates. Essentially, self-shading from the thick meadow is thought to
effectuate light limitation (Binzer et al., 2006; Sand-Jensen, 2007). Light saturation ($I_k$) at the Elba meadows occurred at less than half of peak irradiance. These results are surprisingly similar to light saturation of photosynthesis in *P. oceanica* fragments (Drew, 1979; Pirc, 1986; Figueroa et al., 2002). Unlike in incubations, saturation occurred for a complete meadow under low in situ light conditions. It contrasts with light-limitation of mature *Z. marina* and *T. testudinum* meadows under in situ irradiance (Rheuban et al. 2014; Long et al., 2015), and suggests that a factor other than light may limit primary production at peak irradiance. Nutrient
limitation may contribute. Chlorophyll content and the maximum photosynthetic rate of seagrasses increase with nutrient availability (Agawin et al., 1996; Alcoverro et al., 1998; Lee and Dunton 1999). Further measurements would improve our understanding of this response.

### 5 Summary

Open water techniques, including eddy covariance, allow measurements of productivity that integrate all ecosystem components
under in situ environmental conditions. For seagrass, accurate metabolism measurements may require correction for diel changes in dissolved O$_2$ within meadows. Ecologically, the behavior of *P. oceanica* meadows differed in one fundamental way from expectation – at a CO$_2$ vent productivity was diminished. This may result from the effect of enhanced CO$_2$ on seagrass ecosystem productivity, but it also may result from adverse effects of other vent fluid constituents. In other respects, *P. oceanica* ecosystem GPP, R, and NEM under in situ conditions behaved similarly to measurements of *P. oceanica* productivity alone in prior studies.
These meadows had a high productivity, close to half of GPP, and were far more productive than surrounding sands. This is in agreement with prior studies and supports the assertion that *P. oceanica* is a significant location for biotic carbon storage (Fourqurean et al., 2012). Unlike in other seagrass meadows, photosynthesis did not increase linearly with irradiance during



summer, but instead approached a saturating irradiance. This feature, the low productivity of surrounding sands, and low respiration in seagrass meadows, are all consistent with nutrient limitation in this oligotrophic system. Finally, diel cycles of dissolved oxygen
within a meadow suggest a functional adaptation to nutrient limitation in this environment. Daytime peaks and nighttime lulls in wind speed are characteristic of the region in summer and align well with an ideal scenario for a seagrass meadow. Nutrients remineralized during the night can be retained into the morning hours before hydrodynamic exchange is renewed.

*Acknowledgements.* The authors are thankful to the team of researchers at HYDRA Institute for Marine Sciences on Elba. The
authors are also thankful to G. Eickert-Grötzschel, K. Hohmann, V. Hübner, A. Niclas, I. Schröder, and C. Wigand of the Max Planck Institute for Marine Microbiology Microsensor Group as well as to V. Meyer, P. Färber, and H. Osmers of the Electronics Workshop, I. Laub of the Mechanical Engineering Workshop, and V. Asendorf, A. Nordhausen, and F. Schramm of the Seatech Hall for their development and support of research technology used in this study. We thank Anna Lichtschlag for nutrient analyses. This work was financially supported by the Max Planck Society and the Helmholtz Society, and it received funding from the
European Union's Horizon 2020 Research and Innovation Program (#654462, STEMM-CCS).

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



**Table 1.** Characteristics of the benthic habitats investigated at Elba and Panarea ($CO_2$ vent sites) including the distance from

shore, the mean water depth at the site, the mean water velocity during benthic flux measurements (± s.d.), the mean meadow

height, and the meadow coverage.

| Site | Distance from shore (m) | Water depth (m) | Water velocity (cm s$^{-1}$) | Meadow height (cm) | Meadow coverage (%) |
|---|---|---|---|---|---|
| *Seagrass meadows* | | | | | |
| Open-water | 300 | 13.0 | 1.2 ± 0.7 | 60 | 99 |
| Nearshore | 60 | 13.1 | 2.6 ± 1.3 | 60 | 95 |
| $CO_2$ vent | 80 | 12.8 | 3.8 ± 2.6 | 30 | 100 |
| *Bare sands* | | | | | |
| Nearshore | 100 | 13.1 | 2.3 ± 1.2 | - | |
| $CO_2$ vent | 100 | 16.8 | 2.9 ± 1.7 | - | |

**Table 2.** Benthic fluxes measured at Elba and Panarea in May of 2016 and 2017. Respiration (R), gross primary production

(GPP), and net ecosystem metabolism (NEM) calculated according to Eqs. 1, 2, and 3. n is the number of hour-long averaging

intervals. Errors represent standard error. Because daytime oxygen production is not normally distributed, the error of GPP

was estimated as the standard error of R, determined over one night, normalized to GPP.

| Site | R (mmol m$^{-2}$ d$^{-1}$) | n (h) | GPP (mmol m$^{-2}$ d$^{-1}$) | n (h) | NEM (mmol m$^{-2}$ d$^{-1}$) |
|---|---|---|---|---|---|
| *Seagrass meadows* | | | | | |
| Open-water day 1 | 105.9 ± 9.3 | 10 | 159.1 ± 13.9 | 14 | 53.2 ± 16.7 |
| Open-water day 2 | 72.2 ± 11.6 | 10 | 129.3 ± 20.7 | 14 | 57.1 ± 23.7 |
| Nearshore day 1 | 65.9 ± 17.3 | 10 | 151.1 ± 39.8 | 14 | 85.2 ± 43.4 |
| Nearshore day 2 | 91.8 ± 18.4[a] | 20 | 203.7 ± 40.8 | 14 | 111.9 ± 44.7 |
| $CO_2$ vent | 58.5 ± 10.0[a] | 20 | 67.4 ± 19.6 | 14 | 8.9 ± 22.1 |
| *Bare sands* | | | | | |
| Nearshore | 7.3 ± 1.6 | 9 | 12.2 ± 2.7 | 14 | 4.9 ± 3.1 |
| $CO_2$ vent | 11.0 ± 1.1 | 8 | 14.3 ± 1.4[b] | 7 | 3.3 ± 1.7[b] |

[a] indicates R was calculated from data collected over two, sequential nights. [b] indicates GPP was calculated from fluxes

measured over a partial day.



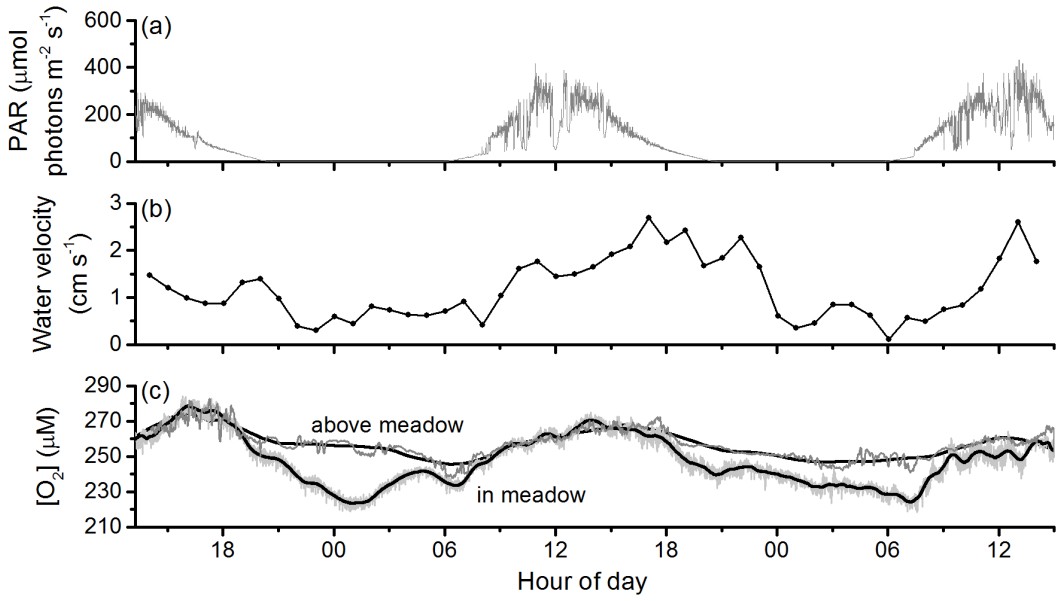

**Figure 1.** Diel changes in photosynthetically active radiation (PAR) **(a),** houry-mean water velocity **(b),** and dissolved oxygen **(c)** at the open-water seagrass meadow in May of 2016. PAR and water velocity were measured above the meadow. Dissolved oxygen is presented at a sampling frequency of once per minute and as a running average. Dissolved oxygen was measured above the meadow and within it.




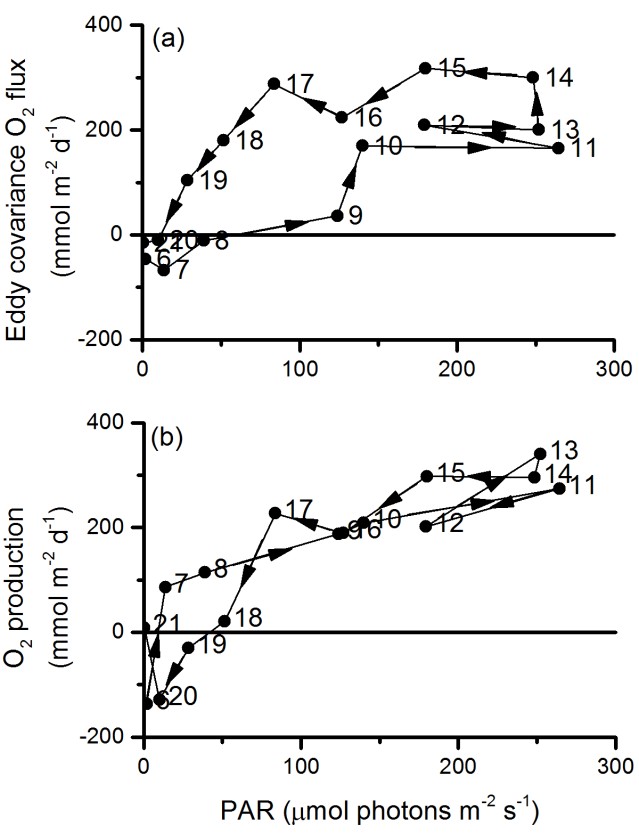

**Figure 2.** Diurnal hysteresis in oxygen flux at the open-water seagrass meadow. Eddy covariance $O_2$ fluxes measured above the meadow as a function of irradiance **(a)**. $O_2$ fluxes corrected to account for $O_2$ concentration change (i.e. storage) in the water between the eddy covariance measurement volume and the sediment, including the meadow **(b)**. Labels represent the hour of the day.





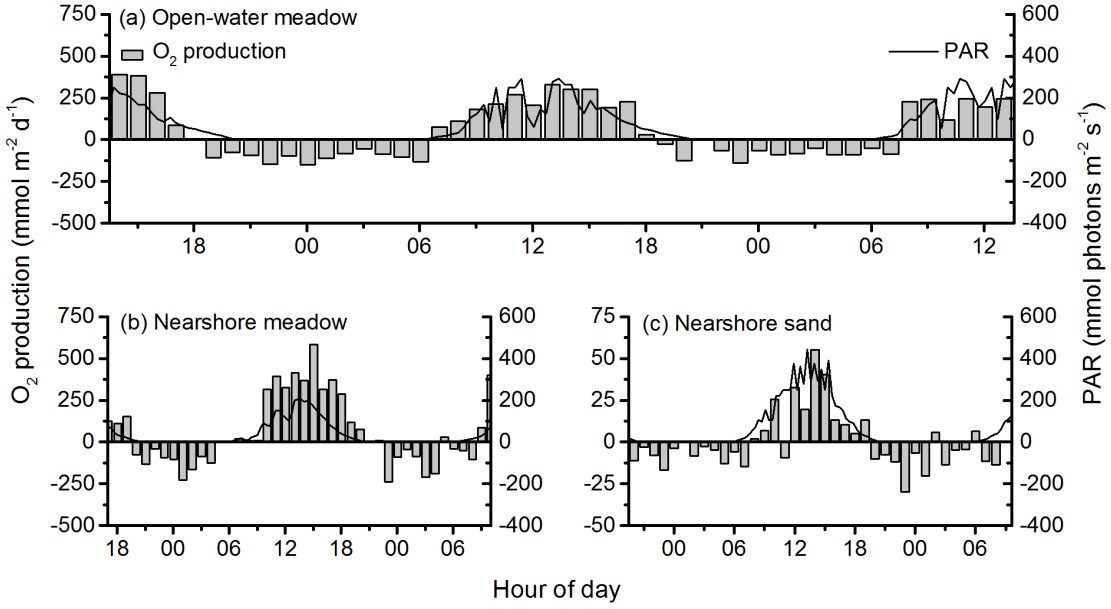

**Figure 3.** Temporal variation in hourly oxygen production (eddy covariance flux + storage) and PAR in the open-water **(a)** and nearshore **(b)** seagrass meadows and bare sands **(c)**. Measurements were made at the island of Elba in May of 2016 and 2017. Note the order of magnitude smaller $O_2$ production in sand.




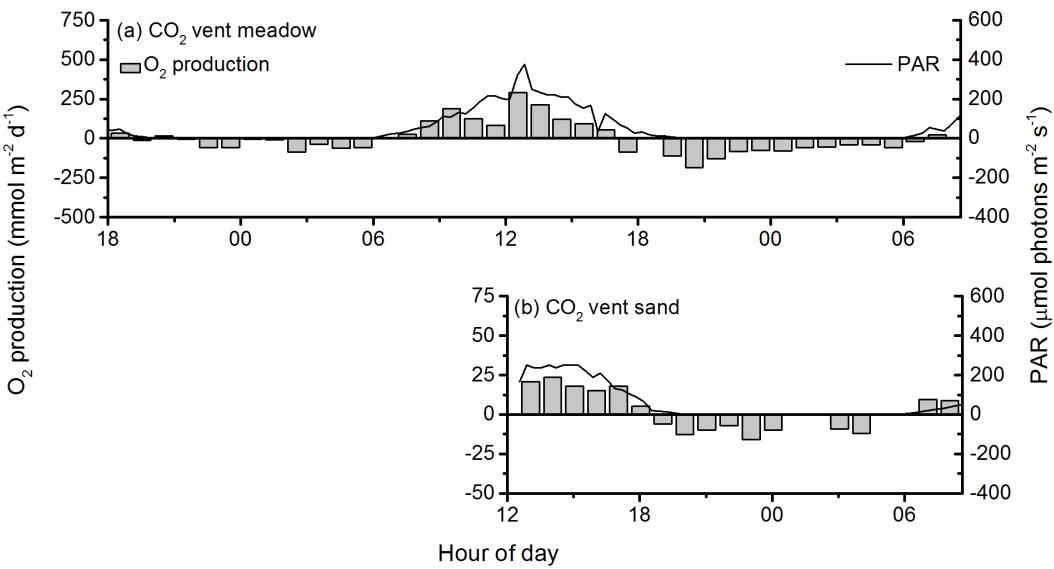

**Figure 4.** Temporal variation in hourly oxygen production (eddy covariance oxygen flux + storage) and PAR in a seagrass meadow **(a)** and over bare sands **(b)** at a $CO_2$ vent at the island of Panarea in May of 2016.





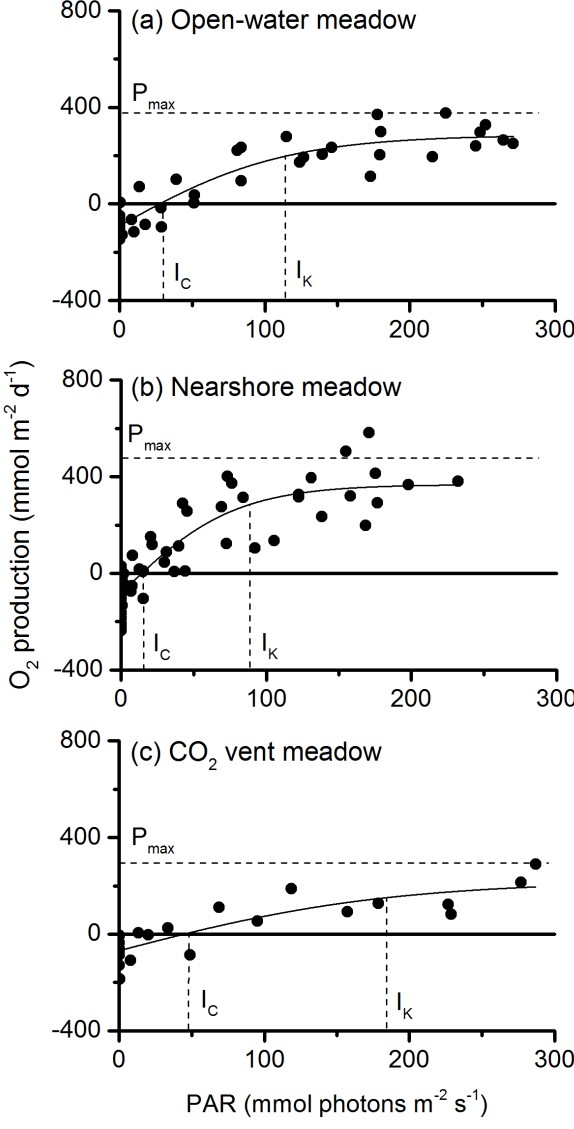

**Figure 5.** Seagrass oxygen production as a function of PAR at open-water **(a)**, nearshore **(b)**, and $CO_2$ vent **(c)** meadows. From **(a)-(c)** the photosynthetic maxima ($P_{max}$) were $375 \pm 27$, $449 \pm 36$ and $289 \pm 75$ mmol m$^{-2}$ d$^{-1}$; light compensation points ($I_C$) were 28, 15, and 49 µmol photons m$^{-2}$ s$^{-1}$; and light saturation parameters ($I_K$) were $113 \pm 21$, $77 \pm 15$, and $185 \pm 98$ µmol photons m$^{-2}$ s$^{-1}$.