# Peer review of "Figure S1. Oxygen production at the open-water seagrass meadow in May of 2016. Time series of PAR (a), dissolved oxygen (b), and eddy covariance oxygen fluxes with- and without accounting for storage of oxygen within the meadow (c)."

_Biogeosciences, 2018_

## Referee Comment (RC1) · Anonymous Referee #1 · 27 Apr 2018

The authors report metabolic measurements in Posidonia oceanica meadows using the technique of eddy-covariance of O2. This is the first publication of such technique in P. oceanica meadow and confirm that this marine community is highly productive as previously shown by numerous studies during the last three decades.

The authors also made primary production of P. oceanica measurements at a CO2 vent site that are compared to a reference site, in the context of ocean acidification. They conclude that the primary production of P. oceanica is lower at the CO2 vent site and attribute this to phosphate binding to iron emitted from the vent. While this interpretation might make sense (although in contradiction with studies of Cymodocea nodosa and other seagrasses at similar CO2 vent sites), I'm uncomfortable about drawing conclusions on the comparison of a single measurement at a single CO2 vent site with a single measurement at a single reference site. It would have been convincing if there was replication of different sites, as there are numerous reasons why two "snapshot" primary production measurements between 2 sites might differ. Carefully replicated measurements might have shown a different result for such a comparison. Even at a single site over a Posidonia oceanica meadow, strong variations of primary production over an order of magnitude occur through a range of temporal scales (from day-to-day to year-to-year).

L261-262: For a meaningful comparison between the CO2 vent site and the reference site, the GPP values should have been normalized by P. oceanica biomass (productivity). Biomass could be lower due other factors such as hydrodynamics (exposed vs sheltered) or substrate or different grazing pressure. Also, biomass strongly changes seasonally and with depth. It is unclear if the measurements in the CO2 vent site and the reference site are really comparable with regards to the timing of the seasonal cycle, and also with regards to other factors affecting primary production such as light availability. Yet on a normalized basis by mass of P. oceanica, the GPP could be equivalent or even higher at the CO2 vent site.

It might be worth mentioning that Cymodocea nodosa biomass (Mishra et al. 2018) and productivity (Apostolaki et al. 2014) seem higher at the vicinity of Mediterranean volcanic CO2 seeps. This seems to also be the case of a variety of seagrass species (Cymodocea serrulata, Cymodocea rotundata, Halodule uninervis, Halophila ovalis, Thalassia hemprichii, and Syringodium isoetifolium) in volcanic CO2 vents in Papua New Guinea (Takahashi et al. 2016)

Minor comments

L161: Specify detection limits, precision and accuracy for nutrient analysis.

L163: Specify the estimated accuracy and precision of TA and DIC measurements.

L165: Do the actual TA measurements show that the values were invariant in time as assumed in the computation of DIC from pH ?

L165: How well did the computed DIC compare to the measured DIC ?

L221: can you put the "importance of seagrass for $CO_2$ sequestration" in perspective with regards to the global carbon cycle ? The global estimate of seagrass net community production is 21-101 TgC/yr as given by the synthesis of Duarte et al. (2010). This number is negligible compared the global anthropogenic $CO_2$ emission of 10,000 TgC/yr (as given by the latest IPCC report).

References

Apostolaki et al. (2014) Seagrass ecosystem response to long-term high CO2 in a Mediterranean volcanic vent, Marine Environmental Research, 99, 9-15, DOI: 10.1016/j.marenvres.2014.05.008

Mishra et al. (2018) Population dynamics of Cymodocea nodosa in the vicinity of volcanic CO2 seeps, in press, www.researchgate.net/publication/323944206_Population_dynamics_of_Cymodocea_nodos

Takahashi et al. (2016) The effects of long-term in situ CO2enrichment on tropical seagrass communities at volcanic vents, ICES Journal of Marine Science, 73, 876–886, doi: 10.1093/icesjms/fsv157

---

## Referee Comment (RC2) · Anonymous Referee #1 · 31 May 2018

I thank the authors for taking into account my suggestions, and for replying in detail to my comments. I would like to react on a couple of points from the reply.

Authors' reply: "If we assume that biomass is proportional to meadow height, (. . .)"

This assumption is only valid if the shoot density is constant. However, this is never the case as shoot density is notoriously variable in time and space (Gobert et al. 2003; Mayot et al. 2006; Terrados & Medina-Pons 2011; Vasapollo & Gambi 2012). Hence, in absence of shoot density data, the authors do not have any grasp on the biomass

differences among the two sites. In absence of this information, the authors cannot conclude if the differences in O2 fluxes are due to different biomass or a response of primary production to the presence of the CO2 vent.

Authors' reply: "the gross primary production normalized for biomass at the CO2 vent is similar to that of the open-water meadow"

This was exactly my point in my initial review. If biomass normalized GPP is similar among sites, then you cannot conclude on nutrient limitation due to the CO2 vent. The conclusion is that changes in pH do not affect the productivity of P. oceanica in line with pH manipulation experiments (Cox et al. 2016).

Authors' reply: "According to Frankignoulle (1986) diel alkalinity changes in seagrass meadows can be 15 mmol L−1, or 0.6% of seawater alkalinity. This small change in alkalinity has an insignificant effect on DIC calculation from pH (Lewis et al., 2008)"

(I assume there's a typo and the alkalinity diel change is 15 $\mu$mol L-1 instead of 15 mmol L-1).

A change of 15 $\mu$mol/L is actually quite a large change in total alkalinity at daily scale that can be assumed to be related calcification from epiphytes and/or dissolution of carbonates, based on Barron et al. (2006). If we assume this, then the related change in DIC is TA/2 (Smith and Key 1975). Based on figures, the observed diel change of O2 was about 40 $\mu$mol/L which translates to a change of DIC of about 30 $\mu$mol/L due to photosynthesis/respiration (O2:DIC = 138:106). Hence the change of alkalinity of 15 $\mu$mol L-1 due calcification/dissolution of carbonates would translate to a change of DIC of 7.5 $\mu$mol/L, equivalent to 24% of the expected change of DIC due to photosynthesis/respiration based on the diel signal of O2 (20 $\mu$mol/L) reported by the authors. This is not an insignificant effect.

Hence, in absence of data to constrain production/dissolution of CaCO3 in the study sites, the authors cannot fully account for "metabolism". They can provide information

on organic carbon metabolism given by O2 fluxes, but leave out the inorganic carbon metabolism has been shown to be relatively important in these ecosystems (Barron et al. 2006), in line with the simple calculations give above.

References

Gobert S et al. 2003 Variations at different spatial scales of Posidonia oceanica (L.) Delile beds; effects on the physico-chemical parameters of the sediment. Oceanologica Acta 26, 199-207

Mayot N et al. 2006 Changes over time of shoot density of the Mediterranean seagrass Posidonia oceanica at its depth limit. Biol. Mar. Medit. 13: 250-254.

Smith, S. V., and G. S. Key (1975), Carbon dioxide and metabolism in marine environments, Limnol. Oceanogr., 20, 493-495.

Terrados J & F J Medina-Pons, 2011, Inter-annual variation of shoot density and biomass, nitrogen and phosphorus content of the leaves, and epiphyte load of the seagrass Posidonia oceanica (L.) Delile off Mallorca, western Mediterranean, Scientia Marina, 75: 61-70

Vasapollo C & MC Gambi, 2012, Spatio-temporal variability in Posidonia oceanica seagrass meadows of the Western Mediterranean: shoot density and plant features, Aquatic Biology, 16: 163-175.

---

## Author Comment (AC1) · 31 May 2018

We thank the referee for their thoughtful consideration of our manuscript and for their valuable suggestions for its improvement. The referee's suggestions are included below in italic.

*The authors report metabolic measurements in Posidonia oceanica meadows using the technique of eddy-covariance of O$_2$. This is the first publication of such technique*

*in P. oceanica meadow and confirm that this marine community is highly productive as previously shown by numerous studies during the last three decades. The authors also made primary production of P. oceanica measurements at a $CO_2$ vent site that are compared to a reference site, in the context of ocean acidification.*

*They conclude that the primary production of P. oceanica is lower at the $CO_2$ vent site and attribute this to phosphate binding to iron emitted from the vent. While this interpretation might make sense (although in contradiction with studies of Cymodocea nodosa and other seagrasses at similar $CO_2$ vent sites), I'm uncomfortable about drawing conclusions on the comparison of a single measurement at a single $CO_2$ vent site with a single measurement at a single reference site.*

*It would have been convincing if there was replication of different sites, as there are numerous reasons why two "snapshot" primary production measurements between 2 sites might differ. Carefully replicated measurements might have shown a different result for such a comparison. Even at a single site over a Posidonia oceanica meadow, strong variations of primary production over an order of magnitude occur through a range of temporal scales (from day-to-day to year-to-year).*

We address the strength of our experimental design, and present ideas for improving its presentation, in four brief sections below.

1. Accounting for diurnal variability

We agree with the referee that even at a single site 'strong variations of primary production over an order of magnitude occur through a range of temporal scales.' These variations are particularly important to address in a study that relies on a single

series of diurnal oxygen flux measurements at an experimental site. We regret that our manuscript does not more clearly convey how we addressed the dominant factors that cause diurnal variability in seagrass primary production. Nor does the manuscript explain the importance of the replication of our measurements at multiple reference sites and over multiple days at each site.

Seagrass primary production is driven primarily by season and irradiance, but also by water temperature, nutrient availability, water velocity, and $CO_2$ concentration (Introduction, lines 26 through 30). All of these factors were accounted for in our experimental design and were measured concurrently with ecosystem metabolism. For example, to address seasonal variation in primary production, all measurements were made during the same two-week period (13 May to 27 May) of the calendar year. To minimize differences in irradiance between meadows, all meadows were at the same water depth.

2. Reproducibility of our results

We also regret that our manuscript does not do a better job of conveying the reproducibility of our results. Unlike flux chamber measurements, eddy covariance measurements of oxygen flux are independent in time. As environmental variables change, the effect of these changes on ecosystem metabolism can be quantified. One-hundred and eighty independent measurements of ecosystem oxygen fluxes were made for this study. As a result, we can report the first ecosystem-scale photosynthesis-irradiance curves for *P. oceanica* meadows.

Ecosystem metabolism measurements were replicated at reference meadows exposed to differing water velocities, and measurements were replicated from day-to-day

at each of the reference meadows. Because of the large footprint of eddy covariance, measurements are comparable to replicated benthic chamber measurements (e.g., Reimers et al., 2012; Berg et al., 2013).

To examine the reproducibility of our results, we made measurements at two different reference meadows and replicated measurements over two days at each of the two meadows. If the factors that drive variability in seagrass primary production were similar from day-to-day, one would expect similar net ecosystem metabolism from day-to-day. We measured changes in the factors that drive primary production and found them to be small. Diel differences in net ecosystem metabolism were also small. The factors that drive primary production were similar at the $CO_2$ vent. The time-of-year, water depth, irradiance, and water temperature were all close to identical to the reference meadows. N-nutrients, $CO_2(g)$, and water velocities were elevated at the $CO_2$ vent. These factors are associated with increases in primary production.

3. Experimental design

Generally speaking, we understand the referee's underlying concern for designing a study based on a single experimental treatment — in this case a $CO_2$ vent — but this design is not unconventional. For example, the referee recommended a highly useful study by Apostolaki et al., (2014) who rely on measurements at a single reference site and a single $CO_2$ vent.

Additional examples cited within this manuscript include Cox et al., (2006) and Barrón et al., (2006). Each study relied on a single experimental treatment. A common factor among these studies is that ecosystem metabolic fluxes were quantified. These require 24 h for measurements of net productivity, reducing the time available for

replication. Other highly influential studies also rely on measurements across carefully selected, but unreplicated sites. For example, Hall-Spencer et al., (2008) observed a doubling of seagrass shoot density at a single, low-pH site.

4. Changes to the manuscript

To address these shortcomings in our manuscript we will make the strength of our experimental design explicit. Specifically, we will explain how we minimized and accounted for the factors that drive diurnal variability in seagrass meadow primary production and respiration, and we will include the evidence from reference meadows that the results are reproducible over time.

To do this, we will present the experimental design as the first section of the methods. In this section we will describe our approach to minimize and account for the factors that drive day-to-day variability in primary production, and our replication of measurements to examine reproducibility over time.

We will also alter Table 2 to include the primary factors that drive seagrass primary production and respiration — cumulative diurnal irradiance, nutrient concentrations, water temperature, $CO_2$ concentration, and water velocity. These will appear alongside seagrass respiration and photosynthesis. The reader can then evaluate whether the factors that drive diel variations in primary production were adequately addressed.

We will also include an evaluation of the reproducibility of measurements in the discussion. This includes a description of the independence of eddy covariance measurements over time and the spatial averaging of the technique. Importantly, this section will discuss our success at minimizing the variability in the factors that

cause diurnal variation in seagrass meadow primary production. It will also include the day-to-day reproducibility of measurements at the reference sites.

*L261-262: For a meaningful comparison between the $CO_2$ vent site and the reference site, the GPP values should have been normalized by P. oceanica biomass (productivity). Biomass could be lower due other factors such as hydrodynamics (exposed vs sheltered) or substrate or different grazing pressure. Also, biomass strongly changes seasonally and with depth. It is unclear if the measurements in the $CO_2$ vent site and the reference site are really comparable with regards to the timing of the seasonal cycle, and also with regards to other factors affecting primary production such as light availability. Yet on a normalized basis by mass of P. oceanica, the GPP could be equivalent or even higher at the $CO_2$ vent site.*

If we assume that biomass is proportional to meadow height, the gross primary production normalized for biomass at the $CO_2$ vent is similar to that of the open-water meadow. However, it would be arbitrary to normalize primary production for biomass without also normalizing respiration. Therefore, net ecosystem metabolism would remain marginal, in stark contrast to the reference sites.

The cause of the short meadow height is directly relevant, and it would be a valuable contribution to the discussion to constrain it. The referee includes hydrodynamics, substrate limitation, grazing pressure, depth, season, and light as drivers of changes in biomass. Our study investigated the effects of hydrodynamics, while controlling for season, depth, and light availability. This leaves substrate limitation, contaminants, and grazing pressure as likely causes of reduced biomass at the $CO_2$ vent. The manuscript discussion includes substrate limitation and contaminants, but we have not addressed grazing. Bite marks are not elevated at the site, but they are present, suggesting that grazing may cause enhanced erosion (Guilini et al., 2017).

*It might be worth mentioning that Cymodocea nodosa biomass (Mishra et al. 2018) and productivity (Apostolaki et al. 2014) seem higher at the vicinity of Mediterranean volcanic $CO_2$ seeps. This seems to also be the case of a variety of seagrass species (Cymodocea serrulata, Cymodocea rotundata, Halodule uninervis, Halophila ovalis, Thalassia hemprichii, and Syringodium isoetifolium) in volcanic $CO_2$ vents in Papua New Guinea (Takahashi et al. 2016).*

Indeed, the study by Apostolaki et al., (2014) is directly relevant to our results. We thank the referee for introducing it to us. This study was conducted at another $CO_2$ vent in the Aeolian islands (Vulcano) where *Cymodocea nodosa* biomass was also reduced but productivity was surprisingly enhanced. These observations were not supported at a $CO_2$ vent in Papua New Guinea where increases in productivity follow increases in biomass at a high $CO_2$ site (Russell et al., 2013). These studies, and the study by Takahashi et al., (2016), are a valuable contribution to our discussion. The study by Mishra et al., (2018) is not yet published.

*Minor comments L161: Specify detection limits, precision and accuracy for nutrient analysis.*

The detection limits of phosphate, nitrate, and ammonium were 0.158, 0.016, and 0.2 $\mu$mol L$^{-1}$, respectively. As an estimate of the precision and accuracy of the technique we will present the standard deviation of replicate measurements of a known concentration. These were 0.018, 0.044, and 0.37 $\mu$mol L$^{-1}$, respectively.

*L163: Specify the estimated accuracy and precision of TA and DIC measurements.*

The precision and accuracy of alkalinity was 8 $\mu$mol L$^{-1}$. The precision and accuracy of DIC was 20 mmol L$^{-1}$. Both estimates are calculated from the standard deviation of replicate measurements. These will be included in the manuscript.

*L165: Do the actual TA measurements show that the values were invariant in time as assumed in the computation of DIC from pH?*

The TA measurements were made at the beginning and end of the deployments, so they do not include diel variation. However, we examined the potential for diurnal variations in alkalinity to affect our calculations of DIC concentration. According to Frankignoulle (1986) diel alkalinity changes in seagrass meadows can be 15 mmol L$^{-1}$, or 0.6% of seawater alkalinity. This small change in alkalinity has an insignificant effect on DIC calculation from pH (Lewis et al., 2008).

*L165: How well did the computed DIC compare to the measured DIC?*

Measured DIC concentrations were 2220 $\pm$ 21 mmol L$^{-1}$ at Elba and 2244 $\pm$ 34 mmol L$^{-1}$ at Panarea (line 94). The mean calculated concentrations were 2215 mmol L$^{-1}$ at Elba and 2220 mmol L$^{-1}$ at Panarea. The high similarity between calculated concentrations at the sites is due to a very similar mean pH.

*L221: Can you put the "importance of seagrass for CO$_2$ sequestration" in perspective with regards to the global carbon cycle ? The global estimate of seagrass net community production is 21-101 TgC/yr as given by the synthesis of Duarte et al. (2010). This number is negligible compared the global anthropogenic CO$_2$ emission of 10,000 TgC/yr (as given by the latest IPCC report).*
This is an important point and we concede it to the referee. It is inaccurate to argue that seagrass carbon sequestration occurs at a rate that is significant compared to anthropogenic emissions. We will revise our statements. Instead, we will make the point that the relatively high rate of carbon storage by seagrasses is locally significant.

References

Apostolaki, E. T., Vizzini, S., Hendriks, I. E. and Olsen, Y. S.: Seagrass ecosystem response to long-term high $CO_2$ in a Mediterranean volcanic vent, Marine environmental research, 99, 9-15, 2014.

Barrón, C., Duarte, C. M., Frankignoulle, M. and Borges, A. V.: Organic carbon metabolism and carbonate dynamics in a Mediterranean seagrass (*Posidonia oceanica*), meadow, Estuaries and Coasts, 29(3), 417-426, 2006.

Berg, P., Long, M. H., Huettel, M., Rheuban, J. E., McGlathery, K. J., Howarth, R. W., Foreman, K. H., Giblin, A. E. and Marino, R.: Eddy correlation measurements of oxygen fluxes in permeable sediments exposed to varying current flow and light, Limnology and Oceanography, 58 (4), 1329-1343, 2013.

Cox, T. E., Gazeau, F., Alliouane, S., Hendriks, I. E., Mahacek, P., Le Fur, A. and Gattuso, J.-P.: Effects of in situ CO2 enrichment on structural characteristics, photosynthesis, and growth of the Mediterranean seagrass *Posidonia oceanica*, Biogeosciences, 13(7), 2179, 2016.

Frankignoulle, M.: Le systéme $CO_2$ en milieu marin: activité biologique, interactions air-mer, caractérisation des masses d´eau dans la couche de surface, in Thése de Doctorat, p. 245, Université de Liege., 1986.

Guilini, K., Weber, M., de Beer, D., Schneider, M., Molari, M., Lott, C., Bodnar, W., Mascart, T., De Troch, M. and Vanreusel, A.: Response of *Posidonia oceanica* seagrass and its epibiont communities to ocean acidification, PloS one, 12(8), e0181531, 2017.

Hall-Spencer, J. M., Rodolfo-Metalpa, R., Martin, S., Ransome, E., Fine, M., Turner, S. M., Rowley, S. J., Tedesco, D. and Buia, M.-C.: Volcanic carbon dioxide vents show ecosystem effects of ocean acidification, Nature, 454(7200), 96, 2008.

Lewis, E., Wallace, D. and Allison, L. J.: Program developed for CO2 system calculations, Carbon Dioxide Information Analysis Center, managed by Lockheed Martin Energy Research Corporation for the US Department of Energy Tennessee., 1998.

Reimers, C. E., Özkan-Haller, H., Berg, P., Devol, A., McCann-Grosvenor, K. and Sanders, R. D.: Benthic oxygen consumption rates during hypoxic conditions on the Oregon continental shelf: Evaluation of the eddy correlation method, Journal of Geophysical Research: Oceans, 117(C2), 2012.

Russell, B. D., Connell, S. D., Uthicke, S., Muehllehner, N., Fabricius, K. E. and Hall-Spencer, J. M.: Future seagrass beds: Can increased productivity lead to increased carbon storage?, Marine Pollution Bulletin, 73(2), 463-469, 2013.

Takahashi, M., Noonan, S. H. C., Fabricius, K. E. and Collier, C. J.: The effects of long-term in situ $CO_2$ enrichment on tropical seagrass communities at volcanic vents, ICES Journal of Marine Science, 73(3), 876-886, 2016.

---

## Referee Comment (RC3) · K. M. Attard (Referee) · 29 Jun 2018

General comments

The paper by Koopmans et al. seeks to address an important scientific question and is within the scope of Biogeosciences. The scientific methods are clearly outlined, and the authors use state-of-the-art methods with clear descriptions of data treatment. Authors give credit to previous work and highlight their own new/original contribution. Overall presentation is well-structured and clear, and the length of the paper is appro-

priate for the dataset. Language is fluent and precise.

The core dataset of the paper consists of benthic oxygen flux measurements. Aquatic eddy covariance (AEC) oxygen fluxes were quantified at 5 shallow sites nearby the Mediterranean islands of Elba and Panarea: three seagrass beds (open-water, nearshore, and CO2 vent) and two sites with bare sands (nearshore and CO2 vent). Flux datasets for the individual sites range in duration from 15 h (CO2 vent bare sands) to 58 h at the nearshore seagrass bed. Based on these datasets, the authors resolve gross primary productivity (GPP), respiration (R), and net ecosystem metabolism (NEM) at each site, and then draw conclusions about metabolism in seagrass and bare sediments in relation to their environmental setting (CO2 levels, hydrodynamic exchange). This dataset is rather limited for the study question being investigated, but I believe there are sufficient novel elements within the data, as well as within the new data processing tools the authors present, to warrant its publication.

Specific comments & technical corrections

Title

Title is clear and reflects the approach taken in the paper. I would suggest adding 'light availability' to the title, e.g. ". . .response of seagrass meadow metabolism to CO2 levels, light availability, and hydrodynamic exchange. . ."

Abstract

L13-15: It would be useful to give some indication of actual rates.

L14: This sentence seems to contradict itself. Perhaps, simply: "Thus, P. oceanica meadows are oases of productivity."

L17: "Oxygen depletion and replenishment within the meadow does not contribute to turbulent O2 flux" This needs to be clarified. Clearly, this process affects the turbulent O2 flux as resolved using the AEC, mostly by 'dampening' the flux signal (Fig. S1). Perhaps: is not captured by turbulent fluxes measured above the canopy?

[Figure]

Methods

L86-87: Study site descriptions. For future studies I would recommend considering biodiversity aspects more carefully. Meadow height and coverage are of interest, but quantifying shoot densities, animal and plant biomass, and presence of ephemeral algae, for instance, would go a long way with helping to better interpret the resolved rates of metabolism.

L87: Please add daily integrated PAR, or daily average PAR to Table 1 or 2. Otherwise it is very difficult to interpret GPP values at the different sites.

L99: Should read "10s of m2"

L132-133: "...do not contribute to fluxes above the meadow". They do, otherwise you wouldn't measure a dampened flux. It is essentially a "missed flux"; a flux that is not captured by AEC measurements above the canopy.

L140: Is there another way to phrase this, instead of 'negative production'? Consumption reflects (secondary) production.

Results

L174-176: How were these incubations performed? Presumably only on parts of the leaves?

L180: Should read "...overlying seawater".

L186-187: Is this referring to photosynthetic production or to net O2 flux? That is, is this difference due to actual decreased photosynthetic production, or is it due to higher photosynthesis-coupled respiration in the afternoon?

L195: GPP values in Table 2. What explains the difference in GPP from one day to the next at the open-water and nearshore seagrass meadows? Light availability, perhaps? It would be informative to have daily integrated PAR values (e.g. in mol photons m-2 d-1) for day 1 and day 2.

L213-214: "In none of the meadows Pmax was reached" needs to be rephrased.

L215: "Ik varied was one-third..." rephrase.

Discussion

L230: Typo- should read "greater"

L234: I suppose that differences in above- vs below-ground biomass, i.e. the ratio between photosynthetic and non-photosynthetic tissue can be different for different species of seagrass (e.g. Duarte and Chiscano 1999 Aquatic Botany). Furthermore, it is important to keep in mind that eddy fluxes represent habitat-scale fluxes, and not just seagrass respiration. Animals, for instance, will contribute through respiration and bioturbation.

L252: "Epifauna biomass..." Presumably autotrophic epiphytes would contribute to the eddy flux signal also?

L258-259: This conclusion is based upon a 'snapshot' dataset. Without investigating this in more detail (e.g. a seasonal study), it may come across as a little premature. It should be stated clearly that these results are specific for the period of investigation.

L265: "...but the negligible NEM suggests that this meadow was not storing organic carbon." It really depends on how production and respiration are partitioned within that habitat. This statement suggests that all of the new production by the seagrass is consumed, but seagrass C:N typically is high, so what is consuming all of that biomass? Presumably these plants are growing and are shedding leaves on an annual basis. There exist other sources of organic matter than the seagrass themselves. One alternative theory could be that seagrass GPP > R, but R is stimulated by sediment entrapment, resulting in a GPP $\approx$ R.

L287: "...hydrodynamic exchange with surrounding waters is limited." Again, this is based on a small dataset, and was not observed at the other sites. 'Can be limited', perhaps?

L325: "These meadows had high productivity..." Is this referring to NEM? If so, this needs to be specified.

L329-332: As I understand it, the point being made here is based upon a single flux dataset (the one that required O2 storage correction). The other datasets did not require this correction, and thus (presumably), this functional adaptation applies only to this one site. However, GPP and NEM rates at the nearshore seagrass sites (no storage) were comparable or higher than the rates observed at the offshore site, which seems contradictory.

Figures

Figure 3: Typo in units for PAR (should be $\mu$mol photons m-2 s-1)

Figure 5: Typo in units for PAR (should be $\mu$mol photons m-2 s-1)

---

## Author Comment (AC2) · 23 Jul 2018

I (DK) would like to apologize to the referee for my delay in submitting this response from all of the authors. We are grateful to the referee for their continued thoughtful contributions to this manuscript. They introduce points that we were also concerned about in the development of this study, and points that we had not considered. We address the rationale for our approach, and introduce improvements that can be made to the manuscript below. The referee's suggestions follow in italic.

[Figure]

*This assumption [that biomass is proportional to meadow height] is only valid if shoot density is constant. However, this is never the case as shoot density is notoriously variable in time and space (Gobert et al. 2003; Mayot et al. 2006; Terrados and Medina-Pons 2011; Vasapollo and Gambi 2012). Hence, in absence of shoot density data, the authors do not have any grasp on the biomass differences among the two sites. In absence of this information, the authors cannot conclude if the differences in $O_2$ fluxes are due to different biomass or a response of primary production to the presence of a $CO_2$ vent.*

We agree that biomass measurements would be a valuable addition to this study. However, as the referee states, shoot density is notoriously variable in time and space. Biomass is even more variable, and the greatest variability is observed at the smallest measurement scale, i.e., quadrats (Vasapollo and Gambi 2012). Because eddy covariance integrates oxygen fluxes over larger spatial scales, a large number of biomass measurements would be required to sufficiently characterize biomass within the footprint. Dive time was limited, so this would reduce the number of eddy deployments.

Our rationale in pursuing eddy covariance deployments to the exclusion of biomass measurements was that the effect of the $CO_2$ vent on seagrass productivity can be revealed from eddy covariance measurements alone. Many prior studies have examined seagrass biomass at $CO_2$ vents (e.g., Hall-Spencer et al. 2008; Apostolaki et al 2014; Takahashi et al. 2016). In these studies, biomass was quantified to examine the effect of the $CO_2$ vent on seagrass net primary production. Our study used oxygen fluxes to quantify net primary production directly. Implicit in the design of the experiment is that the effect on seagrass productivity is due to the $CO_2$ vent. The same implicit assumption was relied on for the above studies.
We agree with the referee in an important respect. We saw that net seagrass meadow primary production was low at the $CO_2$ vent, but without biomass we cannot tell if gross primary production was elevated within leaf tissues at the $CO_2$ vent. We will now include in the discussion the possibility that biomass-normalized gross primary production was elevated. However, in the absence of an understanding of the vertical distribution of photosynthetic production within these meadows, normalizing by biomass may be a mistake. Dalla Via et al., (1998) found that 50-60% of light was attenuated by horizontal fronds at the top of a *P. oceanica* canopy. In *T. testudinum* meadows 34 to 90% of irradiance was attenuated in the top 20 cm (Enriquez and Pantoja-Reyes 2005). Therefore, it is likely that the top of the meadow contributes disproportionately to primary production. To our knowledge, the vertical distribution of photosynthesis has not been resolved in a seagrass meadow. However, as the fraction of downwelling irradiance absorbed by a meadow increases from 20 to 90%, biomass-normalized photosynthesis declines three-fold (Zimmerman 2003). Zimmerman (2003) also found that canopy height is a good predictor of irradiance absorbed. Thus, we would expect biomass-normalized photosynthesis to be greater in the short meadow at the $CO_2$ vent than at the taller meadows at Elba due to differences in canopy architecture alone. In this way, normalizing by biomass may obscure, rather than reveal, the effect of the $CO_2$ vent on seagrass meadow productivity. Instead of normalizing by biomass, our approach was to normalize by area. This approach avoids the confounding effects of canopy architecture on biomass-normalized seagrass meadow primary production.

*This [assumption that if biomass is proportional to meadow height, the gross primary production of the $CO_2$ vent is similar to that of the open-water meadow] was exactly my point in my initial review. If biomass normalized GPP is similar among sites, then you cannot conclude on nutrient limitation due to the CO2 vent. The conclusion is that changes in pH do not affect the productivity of P. oceanica, in line with pH manipulation experiments (Cox et al., 2016).*

We agree with the referee's fundamental suggestion that given the low meadow height, gross primary production may be elevated within leaf tissues at the $CO_2$ vent. We find, however, that nutrient limitation is one of the reasonable explanations for reduced meadow productivity at the vent. Nutrient-limited *P. oceanica* meadows add biomass when nutrients are added (Alcoverro et al., 1997). Between nutrient-limited and nutrient-replete conditions, seagrass biomass may double (Powell et al., 1989). Therefore, biomass-normalized productivity at nutrient-limited and nutrient replete meadows may be confounded by biomass. The dependency of biomass-normalized primary production on canopy height (Zimmerman 2003), makes a comparison of biomass-normalized primary production across these sites potentially misleading. Under these circumstances, area-normalized measurements offer advantages over biomass-normalized measurements for identifying suppressed meadow productivity. We will address this gap in our manuscript by justifying area-normalized measurements in the introduction.

*A change of 15 $\mu$mol L-1 is actually quite a large change in total alkalinity... Hence, in absence of data to constrain production/dissolution of $CaCO_3$ in the study sites, the authors cannot fully account for "metabolism." They can provide information on organic carbon metabolism given by $O_2$ fluxes, but leaving out the inorganic carbon metabolism has been shown to be relatively important in these ecosystems (Barron et al., 2006).*

We thank the referee for making this point and for their calculations to estimate the contribution of inorganic carbon to metabolism. We agree with the referee and regret that we did not address this point in our discussion. Our terminology and assumptions follow convention (e.g., Duarte et al., 2010), but production and dissolution of $CaCO_3$ could indeed enhance total primary production by perhaps one-fourth. We will

include this point, and rationale for the magnitude of this contribution, in our discussion.

References

Alcoverro, T., Romero, J., Duarte, C. M. and López, N. I.: Spatial and temporal variations in nutrient limitation of seagrass Posidonia oceanica growth in the NW Mediterranean, Marine Ecology Progress Series, 155–161, 1997.

Apostolaki, E. T., Vizzini, S., Hendriks, I. E. and Olsen, Y. S.: Seagrass ecosystem response to long-term high $CO_2$ in a Mediterranean volcanic vent, Marine environmental research, 99, 9–15, 2014.

Dalla Via, J., Sturmbauer, C., Schönweger, G., Sötz, E., Mathekowitsch, S., Stifter, M. and Rieger, R.: Light gradients and meadow structure in Posidonia oceanica: ecomorphological and functional correlates, Marine Ecology Progress Series, 163, 267–278, 1998.

Duarte, C. M., Marbà, N., Gacia, E., Fourqurean, J. W., Beggins, J., Barrón, C. and Apostolaki, E. T.: Seagrass community metabolism: Assessing the carbon sink capacity of seagrass meadows, Global Biogeochemical Cycles, 24(4), 2010.

Enríquez, S. and Pantoja-Reyes, N. I.: Form-function analysis of the effect of canopy morphology on leaf self-shading in the seagrass Thalassia testudinum, Oecologia, 145(2), 234–242, 2005.

Hall-Spencer, J. M., Rodolfo-Metalpa, R., Martin, S., Ransome, E., Fine, M., Turner,

S. M., Rowley, S. J., Tedesco, D. and Buia, M.-C.: Volcanic carbon dioxide vents show ecosystem effects of ocean acidification, Nature, 454(7200), 96, 2008.

Powell, G. V., Kenworthy, J. W. and Fourqurean, J. W.: Experimental evidence for nutrient limitation of seagrass growth in a tropical estuary with restricted circulation, Bulletin of Marine Science, 44(1), 324–340, 1989.

Takahashi, M., Noonan, S. H. C., Fabricius, K. E. and Collier, C. J.: The effects of long-term in situ CO2 enrichment on tropical seagrass communities at volcanic vents, ICES Journal of Marine Science, 73(3), 876–886, 2016.

Vasapollo, C. and Gambi, M. C.: Spatio-temporal variability in Posidonia oceanica seagrass meadows of the Western Mediterranean: shoot density and plant features, Aquatic Biology, 16(2), 163–175, 2012.

Zimmerman, R. C.: A biooptical model of irradiance distribution and photosynthesis in seagrass canopies, Limnology and oceanography, 48(1, part 2), 568–585, 2003.

---

## Author Comment (AC3) · 27 Jul 2018

We thank the referee for his careful reading of our manuscript and his thoughtful suggestions for its improvement. He raised points that we had not considered and identified mistakes in a few of our assumptions. His recommendations are listed below in italic. Our responses follow in regular type.

*Title is clear and reflects the approach taken in the paper. I would suggest adding 'light*

*availability' to the title, e.g. ". . .response of seagrass meadow metabolism to $CO_2$ levels, light availability, and hydrodynamic exchange. . ."*

We agree that diurnal light availability is a significant portion of the results of the manuscript. We will change the title to "The response of seagrass (Posidonia oceanica) meadow metabolism to $CO_2$ levels, light availability, and hydrodynamic exchange determined with aquatic eddy covariance."

Abstract
*L13-15: It would be useful to give some indication of actual rates.*

We will alter the abstract to state "Seagrass net ecosystem metabolism was 53 to 112 mmol $m^{-2}$ $d^{-1}$."

*L14: This sentence seems to contradict itself. Perhaps, simply: "Thus, P. oceanica meadows are oases of productivity."*

We understand the contradiction but would like to include a mention of the low productivity of the surrounding area. We would change the statement to "*P. oceanica* meadows are oases of productivity in unproductive surroundings."

*L17: "Oxygen depletion and replenishment within the meadow does not contribute to turbulent $O_2$ flux" This needs to be clarified. Clearly, this process affects the turbulent $O_2$ flux as resolved using the AEC, mostly by 'dampening' the flux signal (Fig. S1). Perhaps: is not captured by turbulent fluxes measured above the canopy?*

We understand your point. We would alter the abstract to state "Oxygen depletion and replenishment within the meadow are not included in turbulent fluxes above the meadow." We respond in more detail at your suggestion to lines 132-133, below.

Methods
*L86-87: Study site descriptions. For future studies I would recommend considering*

[Figure]

*biodiversity aspects more carefully. Meadow height and coverage are of interest, but quantifying shoot densities, animal and plant biomass, and presence of ephemeral algae, for instance, would go a long way with helping to better interpret the resolved rates of metabolism.*

We agree. We also would have liked to include these measurements in our characterizations of the sites and we regret that we did not address these shortcomings in our manuscript. We will include discussions of the advantages of shoot density and biomass measurements. Additionally, macroalgae in particular can enhance primary production and respiration in seagrass meadows (McGlathery 2001). No macroalgae were observed within seagrass meadows at our study sites. Benthic macroalgae was present near the meadows at Panarea, however. We will state this in the study site descriptions.

*L87: Please add daily integrated PAR, or daily average PAR to Table 1 or 2. Otherwise it is very difficult to interpret GPP values at the different sites.*

We agree that adding daily average PAR to one of the tables will help the manuscript. We will include it in Table 2 to show variation from day-to-day. We will also include nutrient concentrations.

*L99: Should read "10s of m$^2$".*

This is an important mistake. Thank you for catching it. The footprint is 10s of meters squared and not 10 m$^2$. We will update this in our copy of the manuscript.

*L132-133: ". . .do not contribute to fluxes above the meadow". They do, otherwise you wouldn't measure a dampened flux. It is essentially a "missed flux"; a flux that is not captured by AEC measurements above the canopy.*

To be more precise we will state that "The diurnal variations in mean O$_2$ concentration within this layer are driven by photosynthesis and respiration, but do not lead immediately to dynamics in fluxes above the meadow. Due to the reservoir in the canopy,

fluxes above the meadows are not instantly (or directly) coupled to the processes in the canopy."

*L140: Is there another way to phrase this, instead of 'negative production'? Consumption reflects (secondary) production.*

We will state that "The net benthic uptake of oxygen represents $O_2$ consumption."

Results

*L174-176: How were these incubations performed? Presumably only on parts of the leaves?*

This was an oversight on my part. We will include the following in the methods. "An oxygen microsensor was prepared and calibrated as described previously (Revsbech 1989). The sensor was mounted on a motorized micromanipulator. Motor control and data acquisition was performed with custom made software (e.g., de Beer and Schramm 1999). Whole seagrass plants, rooted in sediment, were placed in an aquarium. Leaves were attached with rubber bands to a sponge and the microsensor tip was positioned at the leaf surface. While the whole plant was exposed to light-dark dynamics at 370 $\mu$mol photons m$^{-2}$ s$^{-1}$, oxygen dynamics were recorded."

*L180: Should read ". . .overlying seawater".*

We agree and will make the change.

*L186-187: Is this referring to photosynthetic production or to net $O_2$ flux? That is, is this difference due to actual decreased photosynthetic production, or is it due to higher photosynthesis-coupled respiration in the afternoon?*

This is an error on my part. I had stated "photosynthetic production" but you are correct that we cannot distinguish photosynthetic production from photosynthesis-coupled respiration. We will change the text from "photosynthetic production" to "$O_2$ production". Specifically, we will state "After the correction, $O_2$ production in the early morning was

greater than $O_2$ production in the evening at the same light levels..."

*L195: GPP values in Table 2. What explains the difference in GPP from one day to the next at the open-water and nearshore seagrass meadows? Light availability, perhaps? It would be informative to have daily integrated PAR values (e.g. in mol photons $m^{-2}$ $d^{-1}$) for day 1 and day 2.*

We will include mean daily PAR in Table 2 for readers to evaluate for themselves, but briefly, irradiance can explain day-to-day differences in primary production at the nearshore meadow, but it cannot explain differences in primary production at the open-water meadow.

A factor in day-to-day variability that we have not addressed in this manuscript is spatial heterogeneity in the meadows. As flow direction changes, the footprint of the eddy covariance technique will follow. Differences in the abundance of seagrass in different directions from the eddy covariance instruments might explain this variability at the open-water meadow. We will also include this description in the discussion.

*L213-214: "In none of the meadows $P_{max}$ was reached" needs to be rephrased.*

We will change the text to "The maximum photosynthetic rate, $P_{max}$, calculated according to Eq. 6, was not reached in any of the meadows"

*L215: "$I_k$ varied was one-third. . ." rephrase.*

We will change the manuscript so that it states "($I_k$) was one-third of peak irradiance"

Discussion
*L230: Typo- should read "greater".*

Thank you for catching it. We will correct the typo.

*L234: I suppose that differences in above- vs below-ground biomass, i.e. the ratio between photosynthetic and non-photosynthetic tissue can be different for different*

*species of seagrass (e.g. Duarte and Chiscano 1999 Aquatic Botany). Furthermore, it is important to keep in mind that eddy fluxes represent habitat-scale fluxes, and not just seagrass respiration. Animals, for instance, will contribute through respiration and bioturbation.*

We agree that differences in above- vs. below-ground biomass could contribute to differences in metabolism between species. *P. oceanica* has the greatest below-ground biomass of the three species, but surprisingly it has the lowest below-ground productivity (Duarte and Chiscano 1989). Ten percent of fixed carbon is allocated to root growth in *P. oceanica*. Thirty percent of fixed carbon is allocated to root growth in the other species. The low proportion of primary production dedicated to root growth in *P. oceanica* may help keep respiration low. We will add this point to the discussion.

We also agree that other flora and fauna within the meadows may contribute differences in meadow metabolism. Animals, as you suggest, are a good example. There can be up to 50,000 benthic invertebrates in a square meter of *Z. marina* meadow (Bostrom and Bonsdorff 1997). It is also relevant that *P. oceanica* peat can be millennia old (Mateo et al., 1997), therefore few consumers are making a living off of it. We regret that we did not include biological surveys of the study sites. We will include a discussion of these points.

*L252: "Epifauna biomass. . ." Presumably autotrophic epiphytes would contribute to the eddy flux signal also?*

This is another aspect of the habitat which is of general interest but was beyond the scope of our experimental design. We discuss the contribution of autotrophic epiphytes to primary production in the Introduction (lines 57 and 58). They can enhance photosynthetic $O_2$ production of seagrass leaves by up to 50% (Libes 1986). We will also include a description that epiphytes were present in all meadows.

*L258-259: This conclusion is based upon a 'snapshot' dataset. Without investigating this in more detail (e.g. a seasonal study), it may come across as a little premature. It*

*should be stated clearly that these results are specific for the period of investigation.*

We will include the duration of measurements. We will write that "Fluxes were determined over only one day and two nights at the $CO_2$ vent, but despite similarities in the drivers of primary production (Table 2), NEM at the $CO_2$ vent was one-sixth to one-twelfth that of the other meadows (Table 2)."

*L265: ". . .but the negligible NEM suggests that this meadow was not storing organic carbon." It really depends on how production and respiration are partitioned within that habitat. This statement suggests that all of the new production by the seagrass is consumed, but seagrass C:N typically is high, so what is consuming all of that biomass? Presumably these plants are growing and are shedding leaves on an annual basis. There exist other sources of organic matter than the seagrass themselves. One alternative theory could be that seagrass GPP > R, but R is stimulated by sediment entrapment, resulting in a GPP similar to R.*

This is a good question. We will adapt our discussion to address it. To begin, we will clarify in the manuscript that net ecosystem metabolism at the $CO_2$ vent meadow was small. 'Negligible' is less accurate. For context, we will compare seagrass meadow primary production and respiration across species using other eddy covariance studies. Interestingly, the proportion of GPP to R at the $CO_2$ vent is similar to the proportion in *Z. marina* and *T. testudinum* meadows in the Mid-Atlantic Bight and in Florida Bay, respectively. Therefore, the high respiration compared to gross primary production at the $CO_2$ vent may be common in other seagrass species. The respiration of imported organic matter is a possible explanation, as you suggest.

*L287: ". . .hydrodynamic exchange with surrounding waters is limited." Again, this is based on a small dataset, and was not observed at the other sites. 'Can be limited', perhaps?*

We will change the text to state "hydrodynamic exchange with surrounding waters can be limited."

[Figure]

*L325: "These meadows had high productivity. . ." Is this referring to NEM? If so, this needs to be specified.*

We will change the text to read "These meadows had a high net ecosystem productivity"

*L329-332: As I understand it, the point being made here is based upon a single flux dataset (the one that required $O_2$ storage correction). The other datasets did not require this correction, and thus (presumably), this functional adaptation applies only to this one site. However, GPP and NEM rates at the nearshore seagrass sites (no storage) were comparable or higher than the rates observed at the offshore site, which seems contradictory.*

*P. oceanica* meadows are distributed broadly in coastal areas of the Mediterranean up to 40 m depth. Thus, the open-water meadow, where the correction due to storage was needed, may be more representative than the nearshore meadow. Indeed, a resistance to mass transfer in *P. oceanica* meadows may be common. It causes the elevation of nutrients and diel oscillations in pH within *P. oceanica* meadows (Gobert et al., 2002; Hendriks et al., 2014). We thank you for pointing out this gap in our manuscript. We will include these points in the discussion.

We will also alter the text of section 4.3, starting at line 287, to address the effects of hydrodynamic exchange at each of the sites. The text will state that "Generally, hydrodynamic exchange enhances seagrass photosynthetic production by increasing the delivery of $CO_2$ and nutrients and increasing the removal of excess $O_2$ (Koch 1994; Thomas and Cornelisen 2003; Mass et al., 2010). The nearshore meadow was exposed to greater water velocities than the open-water meadow. Consistent with this, the nearshore meadow was also the site of greater primary production. Given the advantages of hydrodynamic exchange for enhancing primary production, it is surprising that the open-water meadow would tolerate a resistance to mass transfer. As an explanation we suggest that reduced hydrodynamic exchange would benefit

seagrass if limiting nutrients that were produced during mineralization at night were retained for primary production during the day."

Figures
*Figure 3: Typo in units for PAR (should be $\mu$mol photons m$^{-2}$ s$^{-1}$) Figure 5: Typo in units for PAR (should be $\mu$mol photons m$^{-2}$ s$^{-1}$)*

We will correct the typos in the figures to $\mu$mol photons m$^{-2}$ s$^{-1}$.

We wish to thank the referee again for his thoughtful contributions to our manuscript.

References
de Beer, D. and Schramm, A.: Micro-environments and mass transfer phenomena in biofilms studied with microsensors, Water Science and Technology, 39(7), 173–178, 1999.

Boström, C. and Bonsdorff, E.: Community structure and spatial variation of benthic invertebrates associated with Zostera marina (L.) beds in the northern Baltic Sea, Journal of Sea Research, 37(1–2), 153–166, 1997.

Duarte, C. M. and Chiscano, C. L.: Seagrass biomass and production: a reassessment, Aquatic botany, 65(1–4), 159–174, 1999.

Gobert, S., Laumont, N. and Bouquegneau, J.-M.: Posidonia oceanica meadow: a low nutrient high chlorophyll (LNHC) system?, BMC ecology, 2(1), 9, 2002.

Hendriks, I. E., Olsen, Y. S., Ramajo, L., Basso, L., Moore, T. S., Howard, J. and Duarte, C. M.: Photosynthetic activity buffers ocean acidification in seagrass meadows, 2014.

Koch, E. W.: Hydrodynamics, diffusion-boundary layers and photosynthesis of the seagrasses Thalassia testudinum and Cymodocea nodosa, Marine Biology, 118(4), 767–776, 1994.

Libes, M.: Productivity-irradiance relationship of Posidonia oceanica and its epiphytes, Aquatic Botany, 26, 285–306, 1986.

Mass, T., Genin, A., Shavit, U., Grinstein, M. and Tchernov, D.: Flow enhances photosynthesis in marine benthic autotrophs by increasing the efflux of oxygen from the organism to the water, Proceedings of the National Academy of Sciences, 107(6), 2527–2531, 2010.

Mateo, M. A., Romero, J., Pérez, M., Littler, M. M. and Littler, D. S.: Dynamics of millenary organic deposits resulting from the growth of the Mediterranean seagrass-Posidonia oceanica, Estuarine, Coastal and Shelf Science, 44(1), 103–110, 1997.

McGlathery, K. J.: Macroalgal blooms contribute to the decline of seagrass in nutrient-enriched coastal waters, Journal of Phycology, 37(4), 453–456, 2001.

Revsbech, N. P.: An oxygen microsensor with a guard cathode, Limnology and Oceanography, 34(2), 474–478, 1989.

Thomas, F. I. and Cornelisen, C. D.: Ammonium uptake by seagrass communities: effects of oscillatory versus unidirectional flow, Marine Ecology Progress Series, 247, 51–57, 2003.
* * *